

# A novel approach to calibrating a photo-acoustic absorption spectrometer using polydisperse absorbing aerosol

Katie Foster[1], Rudra Pokhrel[1], Matthew Burkhart[1], Shane Murphy[1]

[1]Atmospheric Science, University of Wyoming, Laramie, 82071, USA

*Correspondence to*: Shane Murphy (shane.murphy@uwyo.edu)

**Abstract.** A new technique for calibrating photo-acoustic aerosol absorption spectrometers with multiple laser passes in the acoustic cavity (multi-pass PAS) has been developed utilizing polydisperse, highly-absorbing, aerosol. This is the first calibration technique for multi-pass PAS instruments that utilizes particles instead of reactive gases and does not require knowledge of the exact size or refractive index of the absorbing aerosol. In this new method, highly-absorbing materials are

aerosolized into a polydisperse distribution and measured simultaneously with a multi-pass PAS and a cavity attenuated phase shift particulate matter single scattering albedo (CAPS PM$_{SSA}$, Aerodyne Inc.) instrument. The CAPS PM$_{SSA}$ measures the bulk absorption coefficient through the subtraction of the scattering coefficient from the extinction coefficient. While this approach can have significant errors in ambient aerosol, the accuracy and precision of the CAPS PM$_{SSA}$ are high when the measured aerosol has a low SSA and particles are less than 300 nm in size where truncation errors are small. To confirm the

precision and accuracy of this approach, a range of aerosol concentrations were sent to the multi-pass PAS and CAPS PM$_{SSA}$ instruments using three different absorbing substances: Aquadag, Regal Black, and Nigrosin. Six repetitions with each of the three substances produced stable calibrations, with the standard deviation of the measurements being less than 2% at 660nm and less than 5% at 405 nm for all substances. Calibrations were also consistent across the different calibration substances (standard deviation of 2% at 660 nm and 10% at 405 nm) except for Nigrosin at 405 nm. The accuracy of the calibration

approach is dependent on the single scattering albedo (SSA) of the calibration substance, but is roughly 6% for the calibration substances used here, which all have an SSA near 0.4. This calibration technique is easily deployed to the field as it involves no toxic or reactive gases and it does not require generation of a monodisperse aerosol. Advantages to this particle-based calibration technique versus techniques based on ozone or nitrogen dioxide absorption include no reactive losses or impact from carrier gases, and the broad absorption characteristics of the particles which eliminate potentially significant errors in

calibration with small errors in the peak wavelength of the laser light.

## 1 Introduction

Absorbing aerosols represent a significant uncertainty in estimates of global radiative forcing. Black carbon (BC) aerosols, which absorbs at all visible wavelengths (Bond et al., 2013) are emitted into the atmosphere as a byproduct of incomplete combustion of biomass and fossil fuels (Bond and Bergstrom, 2006; Jacobson, 2004, 2010). Brown carbon refers to organic





aerosol that absorbs much more strongly in the high-energy (blue) portion of the visible spectrum than the red (Bahadur et al., 2012; Barnard et al., 2008; Kirchstetter and Thatcher, 2012; McMeeking et al., 2014). Bond et al. (2013) estimated the global top of the atmosphere radiative forcing of BC to be 1.1 [0.17 – 2.1] W/m$^2$, compared to the radiative forcing from $CO_2$ of +1.68 [1.5 – 1.86] W/m$^2$ and $CH_4$ at +0.97 [0.80 – 1.14] W/m$^2$ [2013]. This estimate that BC is the second most radiatively

significant emission does not include the radiative effects of brown carbon, which is potentially a significant category of absorbing aerosol but with larger uncertainty in its optical properties and abundance. Modelling studies indicate that the direct radiative forcing of brown carbon could range up to +0.12 W/m$^2$ or +0.57 W/m$^2$ (Lin et al., 2014; Saleh et al., 2015). Much of the uncertainty stems from the dependence on mixing state, (Brown et al., 2018; Cappa et al., 2012; Feng et al., 2013; Liu et al., 2015) and from a wide range of reported refractive indices (Chakrabarty et al., 2010; Lack et al., 2012a; Nakayama et al.,

2013; Saleh et al., 2013, 2014). The actual radiative forcing of brown carbon is significantly less if it bleaches quickly, but the extent and timeframe of bleaching remain unclear (Forrister et al., 2015; Lee et al., 2014; Liu et al., 2016).

        Given the significance and uncertainty of absorbing aerosol radiative forcing, it is critical to have accurate and unbiased measurements of aerosol absorption. There are several ways to measure aerosol absorption. Most commonly, absorption is measured by filter-based techniques such as the aethalometer (Hansen et al., 1984) or particle soot absorption

photometer (PSAP) (Bond et al., 1999). In these approaches, the attenuation of laser energy due to absorption by aerosols that are captured on a filter is measured, but these techniques are prone to a variety of biases from multiple scattering within the filter itself, variability in backscatter based on the size distribution of the particles, and issues with non-linear responses to loading as the filter becomes saturated (Bond et al., 1999; Collaud Coen et al., 2010; Kondo et al., 2009; Lack et al., 2014; Müller et al., 2011; Weingartner and Schnaiter, 2003). Ultimately the precision of ambient absorption measurements by filter-

based measurements is considered to be roughly 30-35% (Bond et al., 2013). An alternate way to measure absorption is via the difference of extinction and scattering (Wei et al., 2013). The difference method is non-ideal due to small measurement errors in extinction and scattering having large impacts on measured absorption levels when the particles are mostly scattering (single scattering albedo is high) (Singh et al., 2014). Scattering measurements are prone to truncation errors for aerosols larger than ~300 nm (when measuring scattering in the visible) due to the high fraction of forward-scattered light for larger size

parameters (Onasch et al., 2015).

        Photoacoustic spectrometry has emerged as an unbiased and sensitive method for measuring absorption of dry aerosol (Arnott et al., 1999; Lack et al., 2006, 2014; Petzold and Niessner, 1996). The PAS technique is described in detail in the methods section, but a PAS with a single laser pass can be calibrated based on first principles, if the resonant cell area, resonant frequency, quality factor of resonator, and the laser beam power at the resonant frequency are known (Rosencwaig, 1980).

This approach was implemented by Arnott et al. (1999, 2000) and was validated by passing a known concertation of nitrogen dioxide through the instrument. However, the sensitivity of the photoacoustic technique is proportional to the laser power inside the acoustic cell and increased sensitivity can be achieved through implementation of an acoustic cell where the laser passes through the cell many times (Lack et al., 2006). Unfortunately, implementation of a multi-pass cell prevents straightforward calibration of the instrument.



For multi-pass instruments it is difficult (Fischer and Smith, 2018b) or not feasible (Lack et al., 2012b) to know the absolute laser power within the cavity, meaning the first principles approach of Arnott et al. (1999) is not possible. Therefore, another calibration approach must be utilized. Lack et al. (2012b) adopted an approach where ozone enriched air is passed in parallel through a photoacoustic cell and a cavity ringdown cell that is operated at the same wavelength. While this approach

has advantages, such as the ease or forming ozone in-situ, there are also significant drawbacks. These drawbacks include 1.) a very small absorption cross section of ozone at 405 nm wavelength ($1.47e{-}23$ $cm^3$ $molecule^{-1}$ (Axson et al., 2011)) necessitating very high ozone concentrations, the need to exactly match the laser wavelengths of the PAS and CRDS, potential reactions or differential wall loss of the ozone between instruments and an apparent dependence of the calibration on the bath gas with a nitrogen bath-gas yielding incorrect slopes (Fischer and Smith, 2018a). Even when accounting for these known

potential issues with ozone calibration, there remains unresolved discrepancies between different research groups (Bluvshtein et al., 2017; Davies et al., 2018; Fischer and Smith, 2018a). Another option is to calibrate with known concentrations of nitrogen dioxide, instead of ozone, running through both the PAS and CRDS. While this has the advantage of being able to have calibrated tanks of $NO_2$, it carries similar negatives to ozone in that $NO_2$ is toxic and reactive, but additional negatives include that $NO_2$ photolyzes significantly at 405 nm (Jones and Bayes, 1973; Lack et al., 2012b). Given the issues with gas-

phase calibration, it would be desirable to have a particle-based calibration method. In addition to avoiding the issues with reactive gasses, a particle-based calibration would enable detection of particle losses in the system. Calibration using particles has been attempted by several groups to assess the validity of their PAS calibration (*Lack et al.*, 2006, 2014; *Bluvshtein et al.*, 2017; *Fischer and Smith*, 2017). All of these groups generated absorbing particles from nigrosin dye and then size-selected the aerosols with a differential mobility analyzer (DMA). The aim of this approach is to determine the absorption through Mie

theory based on knowledge of the refractive index and size of a monodisperse distribution of spherical particles being sent to the instrument. However, size selection via DMA causes two major issues. First, the concentration of particles is dramatically reduced because only a small fraction of the size distribution passes through the instrument and because uncharged particles area lost. Second, large particles with multiple charges will be passed through the DMA along with the monodisperse particles of interest, which causes significant errors in the calibration because these particles have roughly eight (doubly charged) or

more (triply charged) times the mass of the target particles and absorption is roughly proportional to mass. The only way to accurately account for these multiply charged particles is to be confident there are very few particles in the larger size ranges or to add a second DMA. Adding a second DMA is expensive and operationally difficult and even a small number of multiply charged particles will cause significant errors. In addition to the issues of generating a monodisperse distribution, this approach to particle-based calibration also requires exact knowledge of the real and imaginary refractive index of the calibration

particles. The refractive index of nigrosin dye has been tested by several groups as an aerosol calibration standard, and three different groups have published estimates of the complex refractive index (RI) of nigrosin at 405nm (Bluvshtein et al., 2016; Ugelow et al., 2017; Washenfelder et al., 2013). The three studies differ in the retrieved imaginary RI by 15%, indicating that nigrosin does not have consistent optical properties between different batches and is probably not a good candidate for an absorbing calibration substance. In fact, there has been a desire for a substance that can be atomized, that absorbs in the visible,





and that has a known constant refractive index for several years. Zangmeister and Radney (2018) are currently developing a substance that can be atomized from aqueous solution that would have a constant known refractive index and could eventually be a NIST aerosol absorption standard, but at this point that approach still requires selection of a monodisperse distribution.

This paper presents a novel calibration technique that utilizes polydisperse absorbing aerosol and does not require a

substance with a known refractive index. The technique allows measurement of concentrations spanning from a few Mm$^{-1}$ to several hundred Mm$^{-1}$, gives consistent results for several different substances across many laboratory calibrations, and has also been used successfully in the field.

## 2 Materials and Methods

### 2.1 UW Photoacoustic Absorption Spectrometer

The photoacoustic absorption spectrometer utilized in this study (referred to from here on as the University of Wyoming PAS or UW PAS) is based on the design of Lack et al. (2012b), with identical cell construction, lasers, mirrors, microphones, speakers and analog signal conditioning. Some important differences are that the Lack et al. PAS has five cells, while the UW PAS has four, two cells operating at 660nm wavelength, and two at 405nm. One cell at each wavelength is configured to sample dry air, while the other two are typically plumbed to the outlet of a thermodenuder. For the calibrations presented in

this study the thermodenuded cells are run in a bypass mode and represent a duplicate measurement at each wavelength. The UW PAS has custom data-acquisition software and significantly different vibration isolation and cooling systems than the Lack et al. PAS. While these modifications provide utility and noise suppression, they do not fundamentally alter the operation of the instrument. A brief instrument description follows. Ambient air is pulled through two half-wavelength resonant cells which consist of two cylinders (11 cm in length, 1.9 cm diameter) with quarter-wavelength caps. The primary eigenmode

resonates with antinodes at the center of each cell. One cell is illuminated with laser light while the other is not. The signal from the two cells is subtracted in an attempt to remove background noise. Cells are sealed with anti-reflective coated windows to pass laser light, and outside of this enclosed cell are two cylindrical mirrors rotated 90° out of phase. The front mirror has a 2mm hole to pass a collimated laser beam, and radius of curvature of 430mm, while the back mirror cylindrical radius is 470mm. The mirrors are treated with a dielectric coating to be 99.5% reflective. When appropriately aligned, the end result is

an astigmatic pattern that produces many (theoretically 182) passes of the laser light, with energy lost from scattering off the mirrors and windows at each pass and several potential versions of the astigmatic pattern, each with different numbers of passes. The light loss depends on how clean the system is and system alignment and is the reason quantification of laser power in the cell is nearly impossible without a significant amount of additional instrumentation. The laser power is modulated at the resonant frequency of the cell and interacts with absorbing components of the aerosol in the cell, which heat and expand

the air around them at the frequency of modulation. The resulting acoustic wave is measured by two microphones (Knowles Corp. EK-23132-000) placed at each antinode (one in the with laser light passing through it and one in the dark cell). The subtraction and amplification of the two microphone signals is done on a signal processing board identical to Lack et al.





(2012b), is digitized, and then undergoes a Fourier transform into frequency space. The power at the peak resonant frequency is summed with the power 1 Hz to either side, so that the total signal retained is an integrated area across three points 1 Hz apart. Thus the signal from each PAS cell is referred to as integrated area (IA).

The laser must be modulated at each cells' resonant frequency, which is dependent on temperature and pressure, so a frequency calibration is performed at whatever interval the temperature or pressure might be expected to drift significantly (typically at least every 5 minutes). There is a speaker (Knowles Corp. EP-24075-000) in each cell for this purpose, and a speaker calibration is performed regularly to determine the resonant frequency. This resonant frequency calibration is currently performed in a different way than Lack et al. [2012]. The speaker output is swept over range of resonant frequencies at constant output power and the frequency that gives the maximum integrated area is found. The first calibration is done over a wide

range of frequencies (1640-1370 Hz) then subsequent calibrations are done over ~5 Hz ranges given that the resonant frequency has never been observed to vary by more than this between frequency calibrations.

## 2.2 CAPS PM$_{SSA}$

Aerodyne's CAPS PM$_{SSA}$ instrument combines a cavity attenuated phase shift (CAPS) measurement of extinction with an integrating nephelometer measurement of scattering. The instrument uses a form of cavity-enhanced spectroscopy by which a

square wave modulated light emission from an LED is detected as a phase-shifted signal that can be converted to extinction (Kebabian et al., 2007). At the same time, scattered light from particles in the CAPS cavity is integrated across all angles minus the extreme forward and backward directions. The details of the CAPS PM$_{SSA}$ design, principles of operation, calibration, sensitivity, and measurement uncertainty are presented in (Onasch et al., 2015). The advantage of a single instrument that can measure the single scattering albedo of bulk aerosols in real time is that it minimizes potential sampling

errors and biases between the scattering and extinction measurement. The extinction measurement is absolute (similar to cavity ringdown spectroscopy) and therefore does not require routine calibration. The scattering channel is calibrated by linking it to the extinction measurement through measurements of a purely scattering aerosol, as discussed in Section 3.1. The main difficulty with using a CAPS PM$_{SSA}$ to measure the scattering coefficient of ambient aerosols is the truncation of light scattered from larger particles that tend to have a phase function where a large fraction of light is scattered in the forward direction.

Onasch et al. (2015) calculated the truncation as a function of PSL diameter for 660 nm and 450 nm wavelength instruments, and demonstrated that the truncation only becomes significant for particles larger than 300 nm in diameter.

## 2.3 Generation of absorbing aerosol for PAS calibration

Three different absorbing substances were used in this study: Aquadag, Nigrosin, and Regal Black. All three are commonly used to generate absorbing aerosol for optical measurements or for measurements by the single particle soot photometer (SP2)

(Baumgardner et al., 2012; Gysel et al., 2011; Jordan et al., 2015; McMeeking et al., 2014; Saleh et al., 2013). Aquadag (Lot#ON03616890) is a high-viscosity slurry while Nigrosin (Lot#BCBR0628V) and Regal Black (Batch#400R GP-3901) are solid crystals. For all three substances, the method of generating a solution was the same. The exact concentration of the



solution is not critical because atomized particles will be diluted with particle-free air, but the size distribution is important due to the need to have particles smaller than 300 nm to limit truncation in the scattering channel of the CAPS PM$_{SSA}$. A small amount, a few crystals (solids) or a quarter spatula (slurry), of the given substance is mixed with Milli-Q water (Millipore system SimPak2) and progressively diluted until the size distribution of the atomized aerosols has 99% of the mass below 300 nm. After generating a reasonable solution that has a peak in its number size distribution from 40-70nm, the solutions are sonicated for 15 minutes to ensure they are completely dissolved or well mixed with the water.

Figure 1 shows a schematic of the experimental setup. Absorbing aerosols are generated from the solutions using a constant output atomizer (TSI) fed with particle-free (pulled through a HEPA filter) air at 20 psi. The aerosol is then passed through a silica gel diffusion drier (TSI) and further diluted with particle-free air to achieve the desired dilution. The dilution is varied so that a range of concentrations can be measured. The aerosols are dried a second time with a nafion tube drier (PurmaPure PD-100T) that is a permanent part of the UW PAS inlet before the flow is split to four different instruments – the 4-cell PAS, two different wavelength (450 nm, 660 nm) CAPS PM$_{SSA}$ instruments and a TSI scanning mobility particle analyser (SMPS), which is a combination of a differential mobility analyser (DMA) and condensation particle counter (CPC). The SMPS is set up with a 10:1 ratio of sheath to sample flow, using flow rates of 3 liters per minute (lpm) for sheath and 0.3 lpm sample. The output from the DMA is diluted with 0.7 lpm of filtered air to achieve a 1 lpm flowrate for the CPC.

## 3 Results and Discussion

The overall idea of the calibration method is to calibrate a multi-pass PAS based on the absorption of small, highly absorbing (SSA < 0.5), particles measured by the CAPS PM$_{SSA}$. To ensure accurate results, the performance, accuracy and precision of the CAPS PM$_{SSA}$ measurement of absorption, through extinction minus scattering, must first be verified.

### 3.1 Calibration of the CAPS PM$_{SSA}$ scattering channel

The scattering channel for the CAPS PM$_{SSA}$ is calibrated relative to the extinction, because the extinction does not require calibration. To calibrate the scattering channel, polydisperse ammonium sulfate was atomized, dried, and diluted following the methods described in section 2.3. The mean geometric diameter of the atomized solution was tuned (through dilution of the atomized liquid) to be close to 55nm, with less than 1% of the mass at diameters greater than 300 nm, as verified by the SMPS. The concentration of the purely scattering ammonium sulfate aerosol is varied to achieve extinction coefficient values ranging from ~5 Mm$^{-1}$ to ~600 Mm$^{-1}$. A linear fit to the resulting data gives the relationship between the scattering coefficient and extinction coefficient derived by a particular instrument. Figure 2 shows an example of one calibration. The intercept in all cases is very close to zero, and is not used because baseline corrections with a filter are made at regular intervals automatically by both instruments. In Fig. 2a the 660nm instrument has a ratio of scattering to extinction of 0.9045, so the true scattering coefficient is the reported scattering coefficient divided by this slope. Similarly, in Fig. 2b, the scattering signal must be divided by 1.0423. Across 6 calibrations done in this manner, the 450nm CAPS PM$_{SSA}$ calibration slope was 1.0439 ± 0.0073 (0.7% standard deviation from the mean). For the 660nm instrument, the ratio of scattering to extinction averaged over 6 calibrations





is 0.890 ± 0.018 (2% standard deviation from the mean). The errors in this calibration are included in the error estimate for the accuracy of this calibration technique.

Tests were performed to confirm the accuracy of the extinction measurement in the CAPS PM$_{SSA}$. In these tests PSL of various sizes were sent to the SMPS then the flow from the SMPS was split between a TSI 3010 CPC and the CAPS

PM$_{SSA}$. It was found that the extinction measured by the CAPS PM$_{SSA}$ was within 5% of the extinction calculated based on Mie calculations using the PSL size and the number of particles measured by the CPC. The error between the Mie calculations and the CAPS PM$_{SSA}$ is within what is expected based on the size range stated for the PSL's, the counting accuracy of the CPC, and the stated accuracy of the CAPS PM$_{SSA}$.

## 3.2 Calculation of AAE for each substance

The UW PAS has two cells that operate at a wavelength of 405nm and two that operate at a wavelength of 660nm. While the red-LED CAPS PM$_{SSA}$ instrument also operates at 660 nm, the blue-LED CAPS PM$_{SSA}$ operates at 450 nm, a mismatch with the PAS wavelength. Initially a 405 nm CAPS PM$_{SSA}$ instrument was built, but the 405 nm mirrors rapidly degraded requiring going back to the 450 nm wavelength. We demonstrate here that accurate calibration with the proposed method is feasible even when the instruments are ~50 nm separated in wavelength. This suggests that one could calibrate PAS instruments at

different wavelengths without having to have a CAPS PM$_{SSA}$ instrument to match every PAS wavelength. Currently CAPS-PM$_{SSA}$ are available at 630, 660, 530, and 450 nm.

To account for the wavelength difference between the CAPS PM$_{SSA}$ (450 nm) and the UW PAS (405 nm), the absorption angstrom exponent (AAE) was calculated based on the 660 nm and 450 nm CAPS PM$_{SSA}$ measurements of absorption for each calibration and substance. This calculated AAE can then be used to convert the 450 nm absorption

coefficient measured by the CAPS PM$_{SSA}$ into an estimate of the 405 nm absorption coefficient needed to calibrate the multi-pass PAS. Equation (1) shows how AAE is calculated from the two different wavelength CAPS PM$_{SSA}$ measurements of absorption, and the same relationship is used to convert absorption at 450 nm to absorption at 405 nm.

$$AAE = -\frac{\log\left(\frac{abs_{660}}{abs_{450}}\right)}{\log\left(\frac{660}{450}\right)} \tag{1}$$

The experimental AAE is different for each of the three substances used in this study allowing for an assessment of the accuracy

of utilizing AAE derived from measurements at 660 and 450 nm to convert from absorption at 450 nm to absorption at 405 nm. Six different calibrations were conducted with each substance to assess the stability of the estimated AAE. For Aquadag, the average AAE (+/- 1σ) was 0.3423 +/- 0.0357, for Regal Black it was 1.053 +/- 0.022 and for Nigrosin it was -0.4687 +/- 0.1127. Nigrosin had a much higher standard deviation than either Regal Black or Aquadag, perhaps suggesting that even within a single batch the substance does not have consistent optical properties. Additionally, the Nigrosin tested here yielded

a negative AAE, which is inconsistent with Bluvshtein et al., (2016), who found a slightly positive AAE, a result that is discussed further in Section 3.6. The standard deviation expressed as a percentage for each of the substances is: 2% for Regal Black, 10% for Aquadag, and 24% for Nigrosin. These results suggest that, of these substances, Regal Black may be the best



choice for this calibration technique because it has a very stable AAE and the AAE is close to 1, which is often the assumption made for black carbon (Bergstrom et al., 2002; Lack et al., 2012a; Moosmüller et al., 2009, 2011). Aquadag is also a good choice because, while it has slightly more variability in its AAE, the AAE itself is smaller than that of Regal Black meaning the accuracy of its value is less critical. Nigrosin has been shown to have an index of refraction that significantly varies across

the visible wavelengths (Bluvshtein et al., 2016), making the utilization of AAE for Nigrosin potentially less robust. The error introduced by adjusting absorption measurements from 450 to 405 nm with measured AAE was also assessed. Nigrosin has the largest variation in calculated AAE from the different calibrations, but the difference in absorption at 405nm calculated from the highest AAE (-0.3) to the lowest (-0.6), a factor of two in AAE, is only 3%. This demonstrates that even with significant variation in AAE the calibration method is still robust and this adjustment in wavelength causes minimal error. The

errors introduced by AAE for Regal Black and Aquadag are significantly smaller than that for Nigrosin. Figure 3 shows 1 Hertz data from one of the six calibrations with each substance. The AAE for a given substance is fairly stable but does grow noisy when absorption values are <10 Mm$^{-1}$. This noise is particularly pronounced in panel (c), when small (~5 Mm$^{-1}$ absorption) concentrations of Nigrosin do not produce a stable enough signal in the two CAPS PM$_{SSA}$ instruments to accurately calculate AAE.

**3.3 UW PAS stability**

Before applying the CAPS PM$_{SSA}$ calibration to the UW PAS, the noise level of the UW PAS was assessed by plotting the Allan deviation as a function of time. A similar analysis for the CAPS PM$_{SSA}$ can be found in Onasch et al. (2015). The Allan deviation as a function of averaging time is displayed for each cell in the instrument in Fig. 4. While the behavior of each cell is slightly different, on average across the four cells of the instrument, the 1 second Allan variance is 0.6 Mm$^{-1}$ and it is 0.5

Mm$^{-1}$ for the best performing cell. After 60 seconds of averaging the noise drops nearly an order of magnitude to 0.09 Mm$^{-1}$on average and 0.06 Mm$^{-1}$ in the lowest noise cell.

**3.4 Precision of calibration results**

As described in section 2.3, Nigrosin, Aquadag, and Regal Black were aerosolized and passed to the 4 PAS cells (two 405 nm cells, two 660 nm cells), 2 CAPS PM$_{SSA}$ cells (660 and 450 nm), and an SMPS. The purpose of measuring with an SMPS was

to confirm that only a negligible fraction of the polydisperse aerosol mass was at diameters > 300 nm. Absorption was determined at 450 nm and 660 nm by subtracting the CAPS PM$_{SSA}$ measurement of scattering from CAPS PM$_{SSA}$ measurements of extinction. The absorption at 405 nm, where the PAS operates, was determined via measurements of the AAE as outlined in section 3.3. The resulting calibration slope for each channel across 6 calibrations are shown in Fig. 5 and Table 1. The standard deviation of the 6 different calibration slopes at 660nm have a maximum standard deviation of 1.2% of

the mean (for Regal Black). At 405nm, the variation is larger, with Nigrosin having the largest standard deviation of the mean at 4%.



Variations over the six calibrations using a single substance are quite small and for the 660 nm data variation between the substances is also small. However, there is more variation between the three different substances in the blue, up to a 20% difference between Nigrosin and Aquadag, though the results for Aquadag and Regal Black are within 5% of one another. We hypothesize that the optical properties of Nigrosin may vary in such a way that assuming an AAE between 450 nm and 405 nm may be inappropriate.

### 3.5 Accuracy of calibration

The previous section demonstrated that the calibration method is fairly precise with less than 5% variation in the average of six calibration runs or between the results from different substances (other than Nigrosin at 405 nm, which appears to be an outlier). Next, the accuracy of the method is assessed. The fractional accuracy of the extinction coefficient measured by the CAPS PM$_{SSA}$ is found by Onasch et al. (2015) to be ± 0.05 or 5%.. The fractional error in SSA is reported by Onasch et al. (2015) to be 0.01 or 1%, but we find that it is slightly larger for our instrumental setup at 0.02 or 2%. We derive this slightly larger error in SSA because this was the variability in our six repetitions of the scattering to extinction calibration for the CAPS PM$_{SSA}$. The absorption coefficient can be found from CAPS PM$_{SSA}$ data via equation (2)

$$Absorption = Extinction * (1 - SSA) \tag{2}$$

Given this, the fractional error in the absorption coefficient, $\sigma_{abs}$, is found by adding the fractional errors in extinction and the term (1-SSA) in quadrature. The error in the term (1-SSA) is simply two percent of the SSA because the integer 1 has no error. This yields an equation for the fractional error in the absorption coefficient of

$$\sigma_{abs} = \sqrt{\sigma_{ext}^2 + \left(\frac{0.02*SSA}{1-SSA}\right)^2} \tag{3}$$

This fractional error in absorption is displayed (as a percent) as a function of SSA in Fig. 6. As SSA goes to 0, the error in absorption approaches the 5% limit which is the error in extinction alone, but as SSA approaches 1 the error goes to infinity. The SSA of the three calibration substances in the current study are all close to 0.4, which yields an error of ± 5.2%. The high error above an SSA of ~0.85 is a good indicator of the limits of using the CAPS PM$_{SSA}$ for measuring absorption in ambient conditions, and one of the main motivations for making PAS measurements. This graph also gives guidance into the highest SSA substances that one might consider using to calibrate a multi-pass PAS with the CAPS PM$_{SSA}$. Finally, at very low levels of absorption the errors are not defined by equation (3), but are rather dominated by the detection limits of the CAPS PM$_{SSA}$. Despite this, equation (3) is a good representation of the calibration error for the technique presented here because the slope of the calibration line is controlled by measurements with sufficient absorption that equation (3) is valid (see figure 5).





### 3.6 Mie theory applied to the Nigrosin calibration

The refractive index of Nigrosin dye was derived by Bluvshtein et al. (2016) through ellipsometry at both 405nm (m=1.624 + 0.154i) and 660nm (m=1.812 + 0.246i). To evaluate the accuracy of the current calibration approach in another way other than the error propagation done in the previous section, the theoretical absorption of the polydisperse Nigrosin particles used

during the calibration was calculated from Mie theory and compared to the absorption measured by the CAPS PM$_{SSA}$. Size distributions were measured by the SMPS, and absorption estimates were made for every SMPS scan yielding 3 independent calculations of absorption for every concentration level. We performed this calculation at multiple tested indices of refraction that have been previously published in the literature. Figure 6a shows CAPS absorption measured versus calculated absorption from Mie theory at 660 nm using the RI derived by Bluvshtein et al. (2016) (m=1.812 + 0.2461i) and demonstrates excellent

agreement. The same calculations were done at 405nm and 450 nm using the Bluvshtein et al. (2016) values (m=1.624 + 0.154i at 405nm, m=1.605 + 0.1898i at 450nm), but in this case the ratio of measured CAPS absorption to calculated absorption from Mie theory is ~0.58 for both the 405nm comparison and the 450nm comparison. The RI for that gives agreement between the 450 nm CAPS PM$_{SSA}$ measured absorption and Mie theory, shown in Fig. 6b, is m = 1.6 + 0.105i. This is similar to the result by Liu et al. (2013) of m = 1.61 + 0.12i. The different refractive indices found in the literature at 405 nm for Nigrosin

((Washenfelder et al. 2015) m=1.66 + 0.183i; (Ugelow et al., 2017) m=1.57 + 0.133i) suggest that different batches of Nigrosin have significantly different absorptivity and that Nigrosin may not be a good calibration substance at shorter wavelengths.

### 4. Conclusions

A new calibration method for multi-pass photoacoustic absorption spectrometers that uses polydisperse absorbing aerosol and an accompanying measurement of absorption from the CAPS PM$_{SSA}$ has been presented. This method is demonstrated to be

consistent over repeated trials and across three different aerosol types, namely Aquadag, Regal Black, and Nigrosin. The calibration curve represents the relationship between absorption as measured by the CAPS PM$_{SSA}$ to integrated area as measured by the PAS microphones, and is demonstrated to be linear and spanning 3 orders of magnitude, up to ~600 Mm$^{-1}$ of absorption. The method is found to have an absolute accuracy of approximately ±6%. This aerosol-based method of calibration is simple and easy to utilize in both the laboratory and the field, and does not require size-selection. By using absorbing

particles, we eliminate several potential concerns from gas-phase calibrations using nitrogen dioxide or ozone. Namely, there is no potential for reactive losses and concentrations on the order of 100's of Mm$^{-1}$ are easily and safely attained. Additionally, small differences in wavelength between instruments are of negligible consequence. To accommodate for a wavelength difference between 405 nm PAS cells and the 450 nm CAPS PM$_{SSA}$, we calculate the AAE of each species based on the relationship between 660 nm and 450 nm measured absorption coefficients and apply this to convert the 450 nm absorption

coefficient to the absorption coefficient at 405 nm. This is also shown to have small uncertainties, <3%. We also derive the refractive index of a particular batch of Nigrosin at 450 nm to be m = 1.6 + 0.105i, and confirm a previous result at 660 nm of m = 1.812 + 0.246i.



*Data availability.* All data are available from the corresponding authors upon request.

*Acknowledgements.* The authors thank Ernie Lewis for providing his Mie theory code.

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

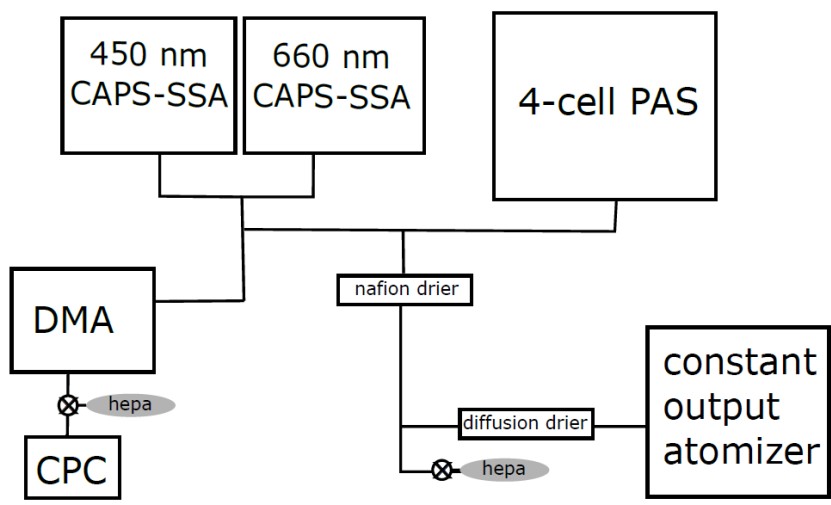



**Figure 1: Schematic of the calibration setup. Flow begins in bottom right at the constant output atomizer, then the aerosols are diluted and dried before being distributed to each instrument. DMA is TSI's differential mobility analyzer, and CPC is the condensation particle counter. When combined, these two instruments are referred to as a scanning mobility particle sizer.**

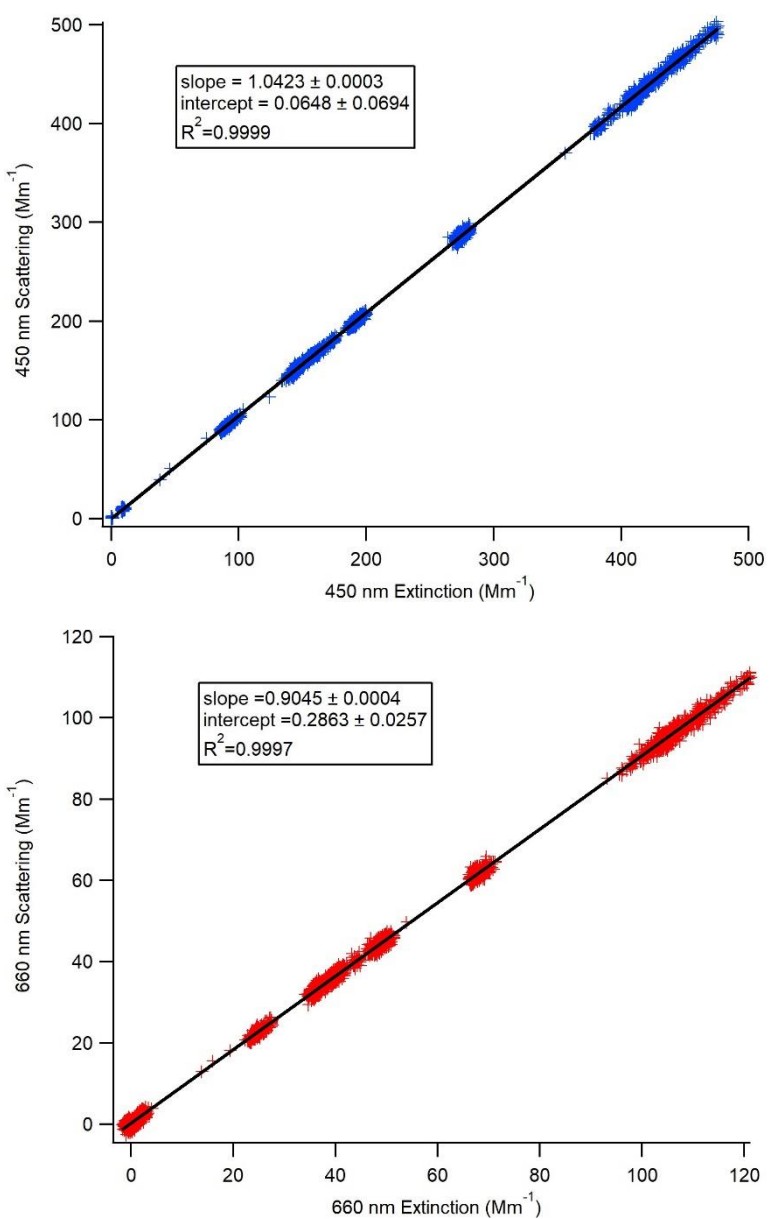

5    **Figure 2: Calibration curves for each of the Aerodyne CAPS PM$_{SSA}$ instruments. The scattering channel is calibrated based on the relationship to the extinction channel across a range of concentrations. The slope of the resulting linear fit gives the ratio that scattering must be corrected by.**








**Figure 3: AAE (black) calculated from 1 Hertz data from the CAPS PM$_{SSA}$ data for three different substances: Aquadag (3a), Regal Black (3b) and Nigrosin (3c). Also shown are 660 nm extinction (black) and 660 nm absorption (red) taken from the CAPS PM$_{SSA}$. The extinction and absorption are shown for context, as the AAE becomes significantly more noisy at low concentrations. Gaps in data occur when the CAPS is conducting a baseline period, and the instruments are switched to filter for several minutes in between each concentration.**

**Figure 4: Allan variance versus averaging time for the UW PAS during filter period. The PAS has 4 cells- 2 at 660nm and 2 and 405nm. During operation one cell at each wavelength can pull denuded air while the other pair of cells measures absorption of dry air.**

| Substance | Average AAE | Average slope at 660nm | Average slope at 405nm |
|---|---|---|---|
| Regal Black | 1.053 ± 0.0216 | 944 ± 12 | 2270 ± 65 |
| Aquadag | 0.342 ± 0.0357 | 961 ± 11 | 2390 ± 33 |
| Nigrosin | -0.469 ± 0.1127 | 921 ± 10 | 1980 ± 75 |

**Table 1: Summary of AAE and clibration slopes for each substance, reported as average and standard deviation from six calibrations.**



**Figure 5: Calibration curves representing CAPS PM$_{SSA}$ absorption versus PAS Integrated Area (IA). The PAS 405 dry cell is on the left, 660 dry on the right. A line is fit to the data, the slope of which gives the relationship between absorption and IA. Intercepts are allowed to achieve the most accurate slopes based on higher absorption levels where the accuracy of the CAPS PM$_{SSA}$ is highest. During operation in the field, both instruments are frequently zeroed based on filter measurements, meaning the intercept of the calibration slope is not needed.**





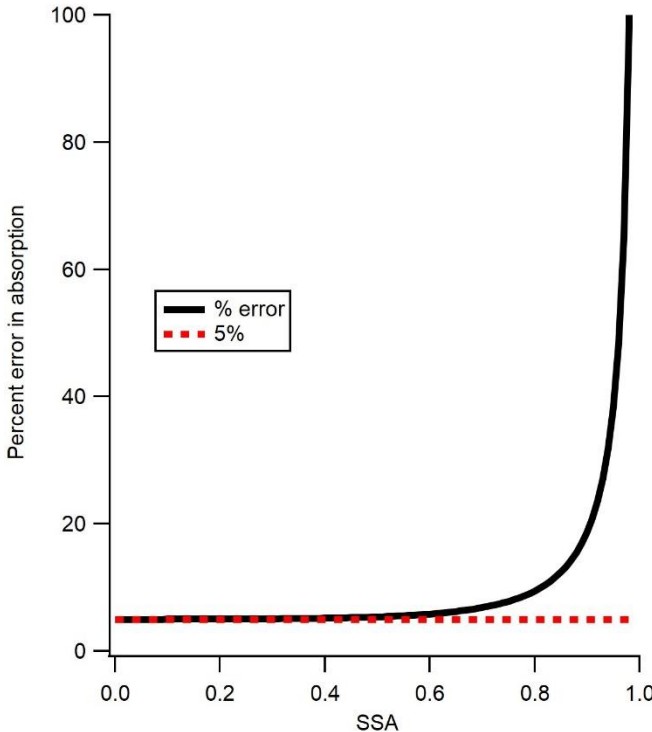

**Figure 6: Percent error in absorption versus single scattering albedo (SSA), following Eq. (3). As SSA approaches zero, absorption error approaches 5%, as SSA approaches 1, error goes to infinity. An SSA of 0.98 has 100% error in absorption.**







**Figure 7: Absorption from CAPS PM$_{SSA}$ data versus calculated absorption from SMPS size distributions, using Mie theory and refractive indices as shown in figure. Panel (a) is from the 660 nm instrument, with calculated absorption using the RI as in Bluvshtein et al. (2017), while panel (b) is from the 450 nm instrument compared to our calculated RI.**

