# Peer review of "A novel approach to calibrating a photo-acoustic absorption spectrometer using polydisperse absorbing aerosol"

_Atmospheric Measurement Techniques, 2018_

## Referee Comment (RC1) · Anonymous Referee #1 · 18 Jan 2019

Manuscript: A Novel Approach to Calibrating a Photo-acoustic Absorption Spectrometer using Polydispersed Absorbing Aerosol (Foster et al.,)

Two technical issues continue to plague the part of our research community focused on better quantifying direct radiative forcing by light absorbing aerosols: measurement bias and calibration standards. First there is the well-documented measurement biases with filter-based measurements, such at the PSAP (particle soot absorption photometer) and the Aethalometer than can lead to overestimation of the aerosol absorption. To get around the use of filter-based measurement and their associated measurement bias photo thermal based measurements (e.g., photoacoustic and other photo thermal

techniques) have been developed. However, this class of instrumentation lacks a broad spectrum calibration standard. The manuscript submitted by Foster et al., describes a technique to address the lack of a broad spectrum photo thermal technique calibration standard by combining simultaneous measurements from their photo acoustic absorption spectrometer and a CAPS-SSA monitor (Aerodyne Research Company instrument that measured aerosol light extinction through the cavity attenuated phase shifted technique and scattering via an incorporated integrating sphere.) By taking the difference between the CAPS-SSA reported extinction and scattering values, aerosol light absorption can be extracted and thus used to calibrate the photo acoustic spectrometer. This approach becomes very useful at shorter visible wavelengths (e.g., 405 nm) where useful gas calibration standards are not available. Given that the work described in this manuscript is of value to only those groups that conduct in situ measurements via the photothermal technique, it is highly myopic and the target audience quite small. However, the subject matter and results are certainly well suited for an AMT venue.

This manuscript has a "draft" feel to it. While the data analysis looks solid there passages in the manuscript that can only be described as sloppy and as such warrants better and more clear writing. I draw your attention to three examples of this. First, the authors cite that "the precision of filter-based measurements is considered to be roughly 30-35%" (page 2, lines 19 & 20). This is wrong (and sloppy) as the PSAP has outstanding precision. Where the PSAP fails is in accuracy as the various correction schemes used to remove measurement bias directly impact the accuracy of the measurement - not its precision. Accuracy is what Bond et al., (reference cited by the authors) also refer to. Second, when discussing the PSAP the authors make reference to the "attenuation of laser energy" (page 2, line 15). This is wrong as the PSAP does not have a laser (it uses LEDs). And third, the statement written on page 5 lines 28-30 "Three different absorbing substances were used in this study: Aquadag, Nigrosin, and Regal Black. All three are commonly used to generate absorbing aerosol for optical measurements or for measurements by the single particle soot photometer

(SP2)". This is very misleading, for one could easily infer that nigrosin is used with the SP2, which is certainly not the case. Errors of this type along with the sloppy writing has cast a shadow over the entire manuscript, which contains interesting results but that are overshadowed. Therefore, this reviewer recommends that this manuscript be rejected so that the authors go over this manuscript more carefully and thoroughly and then resubmit it for publication - which this reviewer believes will find acceptance.

A couple of additional comments.

While this reviewer agrees with the authors stated toxicity concerns of NO2 based calibration, it should be noted that typical concentrations used to calibrate this class of instrumentation are in the 10s - low 100s of ppb range - a range that is easily and safely used in laboratory and field conditions. The big issue for NO2 is our uncertainty with respect to photodissociation at 405 nm and thereby limiting the utility of this gas at this wavelength. In contrast, this gas standard remains a very good (best?) calibration at 532 nm.

The 405 nm calibration curves shown in Figure 5 for both Regal black and Aquadag give intercepts of 11.24 Mmˆ-1 and 5.5 Mmˆ-1, respectively. Do the authors have an explanation for non-zero intercepts? In the atmospherically-relevant range for aerosol absorption (0-25 Mmˆ-1) such an offset is huge. It is interesting to note that the intercept for nigrosin is < 1 Mmˆ-1. A similar trend is seen at 660nm.

This reviewer would like the authors to provide some cautionary text regarding extrapolation of measurements/data collected at 450 nm (CAPS) to 405 nm (PAS). Yes, the results seem to suggest all is fine, but this may be a fortuitous and be a unique case. Indeed, as the authors point out, the standard deviation for nigrosin is significantly larger (3-5x) then the other two calibration standards examined. This larger uncertainty for nigrosin could be due to manufacturers mixing the polyaniline nigrosin pigment (which is bluish/black color) with an orange dye in order to achieve a specific color index (CI: 50420) which could lead to a very different wavelength dependence that is captured

between the CAPS wavelengths and that could, in turn, easily cause an error at the extrapolated wavelength of 405 nm.

The authors may consider putting several of the figures in a supplemental section and limit main figures to those that are most germane to the manuscript subject matter (e.g., figures 2, 5 and 7).

It would be nice to see actual aerosol size distributions for the samples used in these experiments (page 6, lines 1-6). This is a figure that could be shown in the aforementioned supplemental section.

Similarly, this reviewer would also like to have seen some SSA plots from the actual calibration materials used and currently limited to the pure scattering experiments.

---

## Referee Comment (RC2) · Anonymous Referee #2 · 23 Jan 2019

Review of Foster et al., "A novel approach to calibrating a photo-acoustic absorption spectrometer using polydisperse absorbing aerosol"

Foster et al. presents a calibration method for an aerosol absorption photoacoustic spectrometer. This is important work that demonstrates the accuracy of photoacoustic measurements that will then be used for future analyses. This topic is specifically interesting in light of results of Bluvshtein et al. who found that ozone calibration were inaccurate by factor of 2 and the subsequent reports of the calibration of multipass photoacoustic spectrometers that found the ozone calibration to be accurate.

This topic is appropriate for AMT and should be published after the corrections listed

below are addressed.

1) Typical analyses of photoacoustic data have included the effect of the acoustic frequency, quality factor of the acoustic cavity, and laser power. Even if these quantities are constant for these experiments, the authors should include them in their formalism. Generally some more formalism such as the photoacoustic equation relating microphone signal to absorption coefficient would be a benefit. 2) The typical style is to leave a space between quantity and the unit. Please check this, in particular when wavelengths are reported. 3) Some values and errors are reported with too many significant digits

Specific Comments: P3L1: The optical power in these multiplass cells is not unknowable. It can be determined with few simple measurements. 1) Measure the transmission of the rear mirror. 2) place a calibrated optical power meter to measure the optical power leaking through the mirror. 3) account for the mirror transmission and a factor two for a similar amount of light leak through the front mirror to get the optical power in the acoustic cavity. The issue with a fundamental calibration is that the overlap integral of the laser, acoustic mode, and aerosol is not known accurately enough for calibrations. Or possibly the microphone sensitivity and/or the laser power are not known accurately enough for calibration purposes. P3L23: 'area' should be 'are' P4L11: extra space before 'Lack' P4L19: Because this acoustic cavity consists of two high coupled resonators, there are two "primary" eigenmodes. In one mode the pressures at center of both resonators are in-phase and in the other the pressures are 180 deg. out of phase. Please rewrite the sentence to clarify which mode is being used. P5L4: either "each cell's resonant frequency" or "each cells' resonant frequencies" would be appropriate. P5L5: please replace "whatever interval" with a more formal phrase P5L6: The microphone part number does not need to be repeated here. P5L15: track changes indicator on this line P6L2-6: If known please state the concentration of solution that is used. Even if it is not critical, it is a good starting point for future projects and replication. P6L21: in this sentence please replace the 'extinction' with 'extinction channel'

P7L25-30: Nigrosin has a complex absorption spectrum and is not appropriately modeled with an angstrom exponent model. P8L17: Use of the Allan deviation is common in the atmospheric community where it is implicitly presented as a detection limit as a function of averaging time. The Allan deviation is useful for identifying drifts, but detection limits and instrument stability are more accurately characterized using the standard deviation which can also be presented as a function of averaging time. I suggest the authors use the standard deviation instead of the Allan deviation. P9L10: Please remove one of the periods. P9L6: This manuscript should assess the overall measurement accuracy for ambient measurements. This section concludes that the accuracy of these calibrations is roughly +/- 6%. Is this the expected overall accuracy for ambient measurements? If yes, please state that explicitly. If no, please explain why. P10L23: please replace '3' with 'three' P11L4: There seem to be a few problems with some of the references. I suggest the authors look over all them carefully. P11L24: Both AMT and AMTD versions of Bluvshtein are in the reference list P12L2: "K??rcher" P12L21: no journal listed P13L19&22: Initials for Lack should capitalized P13L31: remove "(Julie)" P15L21: No Journal listed Figure 1: Could you re-arrange this figure so the flow goes from left to right. Maybe add arrows. Figure 2: Please change the vertical axis to 'nominal scattering' to indicate it is uncalibrated.

---

## Author Comment (AC1) · 12 Mar 2019

Response to Anonymous Referee #1 comments (RC1) on

*A novel approach to calibrating a photo-acoustic absorption spectrometer using polydisperse absorbing aerosol (https://www.atmos-meas-tech-discuss.net/amt-2018-413/)*

We found the comments from referee #1 to be highly valuable and the paper has been improved because of this diligent review. We have carefully edited the manuscript and we believe the "sloppy" or "draft" feel of the manuscript has been corrected. We believe, given that Anonymous Referee 1 stated that the data analysis "looks solid", that there is not a need for re-submission, but rather that the significantly corrected/improved manuscript submitted to the editor is ready for publication in AMT.

Please note that page and line numbers listed below refer to the locations in the original manuscript posted in AMTD.

Comment #1:
*This manuscript has a "draft" feel to it. While the data analysis looks solid there are passages in the manuscript that can only be described as sloppy and as such warrants better and more clear writing. I draw your attention to three examples of this. First, the authors cite that "the precision of filter-based measurements is considered to be roughly 30-35%" (page 2, lines 19 & 20). This is wrong (and sloppy) as the PSAP has outstanding precision. Where the PSAP fails is in accuracy as the various correction schemes used to remove measurement bias directly impact the accuracy of the measurement - not its precision. Accuracy is what Bond et al., (reference cited by the authors) also refer to.*

Response to comment #1: We are grateful to the referee for pointing out these errors in the language used. We have gone through the manuscript carefully to ensure there are no further "sloppy" errors and have made a number of changes. Specific to this comment, we have replaced "precision" with "accuracy".

P2,L19 now reads: "While filter-based measurements can have high precision, absorption measurements made by filter-based measurements are typically only accurate to within roughly 30-35% (Bond et al., 2013)."

Comment #2:
*Second, when discussing the PSAP the authors make reference to the "attenuation of laser energy" (page 2, line 15). This is wrong as the PSAP does not have a laser (it uses LEDs).*

Response to comment #2: This was indeed a sloppy typo to reference the laser attenuation of a PSAP, when in fact the PSAP uses LEDs.

This sentence now reads (P2,L15) "These approaches utilize a measurement of the attenuation of light intensity (typically from an LED) due to absorption by aerosols that are captured on a filter, but these techniques are prone to a variety of biases from multiple scattering within the filter itself, variability in backscatter based on the size distribution of the particles, and issues with non-linear responses to loading as the filter becomes saturated (Bond et al., 1999; Collaud Coen et al., 2010; Kondo et al., 2009; Lack et al., 2014; Müller et al., 2011; Weingartner et al., 2003)."

Comment #3:
*The statement written on page 5 lines 28-30 "Three different absorbing substances were used in this study: Aquadag, Nigrosin, and Regal Black. All three are commonly used to generate absorbing aerosol for optical measurements or for measurements by the single particle soot photometer (SP2)". This is very misleading, for one could easily infer that nigrosin is used with the SP2, which is certainly not the case.*

Response to comment #3: We agree that the sentence was, unintentionally, misleading and we agree that the SP2 generally uses Aquadag or Fullerene Soot, and not Nigrosin or Regal Black, to calibrate. This part of the paper has been augmented now to be accurate by stating that the three substances are commonly used to generate absorbing aerosol for optical measurements by photoacoustic absorption spectrometers, and Aquadag is commonly used for measurements by the single particle soot photometer (SP2).

New sentence beginning P5,L28: "All three are commonly used to generate absorbing aerosol for optical measurements by photoacoustic absorption spectrometers, and Aquadag is commonly used for measurements by the single particle soot photometer (SP2) (Baumgardner et al., 2012; Gysel et al., 2011; Jordan et al., 2015; McMeeking et al., 2014; Saleh et al., 2013)."

Comment #4:
*While this reviewer agrees with the authors stated toxicity concerns of NO2 based calibration, it should be noted that typical concentrations used to calibrate this class of instrumentation are in the 10s - low 100s of ppb range - a range that is easily and safely used in laboratory and field conditions. The big issue for NO2 is our uncertainty with respect to photodissociation at 405 nm and thereby limiting the utility of this gas at this wavelength. In contrast, this gas standard remains a very good (best?) calibration at 532 nm.*

Response to comment #4: The authors appreciate this input and clarification from the reviewer. The paragraph the reviewer is referring to has now been improved because of this comment.

P3,L12 now reads: "The primary problem with using $NO_2$ to calibrate is that $NO_2$ photolyzes at 405 nm and the magnitude of photolysis depends on the laser power in the instrument (Jones and Bayes, 1973; Lack et al., 2012a), so while it would be a good calibration standard at 532 nm, it is a poor standard near or below 405 nm. Even for 532 nm cells, calibration with $NO_2$ requires exact matching of laser wavelengths between the PAS and CRDS, has the potential for reactive loss, and requires the use of a toxic substance. While the $NO_2$ concentrations are often small enough not to pose a significant health hazard, $NO_2$ use on airborne platforms still requires significant additional safety precautions."

Comment #5:
*The 405 nm calibration curves shown in Figure 5 for both Regal black and Aquadag give intercepts of 11.24 Mmˆ-1 and 5.5 Mmˆ-1, respectively. Do the authors have an explanation for non-zero intercepts? In the atmospherically-relevant range for aerosol absorption (0-25 Mmˆ-1) such an offset is huge. It is interesting to note that the intercept for nigrosin is < 1 Mmˆ-1. A similar trend is seen at 660nm.*

Response to comment #5: The authors thank the reviewer for raising this question, and the matter has been revisited and carefully considered. The calibration method uses 5 different concentrations, and for the regal black (the substance with the largest intercept), the largest concentration is 600 Mm$^{-1}$, while the smallest is 36 Mm$^{-1}$. The slope is therefore calculated with the accuracy stated in the paper, but the intercept is not used as part of the calibration and can be significantly off because the larger concentrations control the slope. To apply the calibration in field or laboratory measurements, we use a filter period to determine zero and then apply our slope from the most recent calibration to signal above this zero level. Were we to include filter data or low concentration points, the intercept would be shifted down to near the origin. In fact, forcing the line to go through the origin alters the slope by only 3% in this particular case. We feel it is best not to force the fit through zero to obtain the best slope, but the slope would not shift dramatically regardless.

We have added the following sentences at P8,L31: "In practice, the calibration slopes are applied to the PAS microphone signal to convert from integrated area to absorption (as outlined in section 2.1). Filter

periods are frequently conducted to determines the background absorption and the PAS data is zeroed to this background. Large (on the order of several hundred $Mm^{-1}$) concentrations are used to generate a slope that can be applied over significant concentrations in field measurements of smoke particles. Therefore, intercepts can be on the order of 10 $Mm^{-1}$. The intercepts from the calibrations are not used."

Comment #6:
*This reviewer would like the authors to provide some cautionary text regarding extrapolation of measurements/data collected at 450 nm (CAPS) to 405 nm (PAS). Yes, the results seem to suggest all is fine, but this may be a fortuitous and be a unique case. Indeed, as the authors point out, the standard deviation for nigrosin is significantly larger (3-5x) then the other two calibration standards examined. This larger uncertainty for nigrosin could be due to manufacturers mixing the polyaniline nigrosin pigment (which is bluish/black color) with an orange dye in order to achieve a specific color index (CI: 50420) which could lead to a very different wavelength dependence that is captured between the CAPS wavelengths and that could, in turn, easily cause an error at the extrapolated wavelength of 405 nm.*

Response to comment #6:
We agree that the method of using absorption angstrom exponent to relate absorption at 450 nm to absorption at 405 nm can be problematic for certain substances. Based on this comment and others from the reviewers, we have expanded Section 3.2 to make the case for using regal black or aquadag rather than nigrosin, given that nigrosin's absorptivity has a complicated dependence on wavelength. We note, however, that even for Nigrosin, shifting absorption from 450 nm and 405 nm via AAE introduces only a 3% bias.

P8,L4 has been adjusted and now reads: "Nigrosin has been shown to have an index of refraction that varies across the visible wavelengths (Bluvshtein et al., 2017), and does not have a relationship between absorption and wavelength that is perfectly modeled by AAE. However, given that the adjustment is only over a small wavelength range, the error introduced by adjusting absorption measurements from 450 to 405 nm with the AAE technique is assessed here."

Comment #7:
*The authors may consider putting several of the figures in a supplemental section and limit main figures to those that are most germane to the manuscript subject matter (e.g., figures 2, 5 and 7).*

Response to comment #7: While we appreciate the desire for brevity, after considering the matter and noting that this is a technique paper, we believe it's appropriate to leave the other figures that show the experimental setup, an example of raw data, instrument noise levels and error analysis. Accordingly, we have decided to maintain the current structure of the paper and keep all figures in the main body. A supplement has been created to include the figures requested below in comments 8 and 9.

Comment #8:
*It would be nice to see actual aerosol size distributions for the samples used in these experiments (page 6, lines 1-6). This is a figure that could be shown in the aforementioned supplemental section.*

Response to comment #8: Aerosol size distributions corresponding to the data displayed in figures 3 and 5 has been provided in SI figure 1. The new supplementary material is provided at the end of this document.

Comment #9:
*This reviewer would also like to have seen some SSA plots from the actual calibration materials used and currently limited to the pure scattering experiments.*

Response to comment #9: SSA data from the experiments shown in figures 3 and 5 can now be found in SI Figure 2.

Table S1: Noise levels of the PAS taken during filter period. Reported is the standard deviation of the mean as a function of averaging time for each of the four PAS cells.

| cell | 1 second data ($Mm^{-1}$) | 30 second average ($Mm^{-1}$) | 60 second average ($Mm^{-1}$) |
|---|---|---|---|
| 405 dry | .0092 | .0039 | .0035 |
| 660 dry | .0687 | .0321 | .0308 |
| 405 den | .0250 | .0311 | .0225 |
| 660 den | .1160 | .0196 | .0199 |

Figure S1: Aerosol size distributions from the three different substances used for the calibration method: Aquadag (a), Regal Black (b), and Nigrosin (c). These size distributions correspond to the example calibration shown in Figures 3, 5, and S2.

[Figure]

Figure S2: Single scattering albedo at 450 nm (blue) and 660 nm (red) for the three substances: Aquadag (a), Regal Black (b), and Nigrosin (c). All three of these examples correspond to the same data used in figures 3, 5, and S1. Also shown is the Extinction at 450 nm (black). The concentration is varied over the course of the calibration, and the lowest concentrations correspond to the highest noise in SSA calculation. For example, in the bottom panel, the first concentration of Nigrosin corresponds to only 7 Mm$^{-1}$ of Extinction at 450 nm, and 3-4 Mm$^{-1}$ of scattering, resulting in a highly noise estimate of

[Figure]

---

## Author Comment (AC2) · 12 Mar 2019

Response to anonymous referee #2 comments (RC2) on

*A novel approach to calibrating a photo-acoustic absorption spectrometer using polydisperse absorbing aerosol (https://www.atmos-meas-tech-discuss.net/amt-2018-413/)*

The authors greatly appreciate the comments, corrections, and suggestions from this anonymous referee. Please note that the individual responses for each comment below use the page and line numbers from the manuscript that was originally submitted in AMTD.

Comment #1:
*Typical analyses of photoacoustic data have included the effect of the acoustic frequency, quality factor of the acoustic cavity, and laser power. Even if these quantities are constant for these experiments, the authors should include them in their formalism. Generally some more formalism such as the photoacoustic equation relating microphone signal to absorption coefficient would be a benefit.*

Response to comment #1:
In the portion of the introduction that discusses photoacoustic spectrometry history and the different calibration approaches, the authors have added a discussion of the photoacoustic equation and the equation itself. These additions are

P3L1: "Theoretically, the absorption ($b_{abs}$) coefficient can be determined from a PAS as a function of absolute laser power ($P_{Laser}$), pressure at the microphone ($P_{Mic}$), resonator cross sectional area ($A_{Res}$), resonant frequency $F_R$, and quality factor (Q).

$$b_{abs} = \frac{P_{Mic}}{P_{Laser}} \frac{A_{Res}}{\gamma - 1} \frac{\pi^2 F_R}{Q} \tag{1}$$

For multi-pass instruments it is difficult (Fischer and Smith, 2018b) or not feasible given the instrument setup (Lack et al., 2012b) to know all of these terms accurately. This means the first principles approach of Arnott et al. (1999) is not possible for many instruments. The issue with a fundamental calibration is that the overlap integral of the laser, acoustic mode, and aerosol is not known accurately enough for calibrations. Additionally, the microphone sensitivity and laser power are not known accurately enough for calibration purposes in the design of Lack et al. (2012b)."

Comment #2:
*The typical style is to leave a space between quantity and the unit. Please check this, in particular when wavelengths are reported.*

Response to comment #2: The authors apologize for this error and appreciate the attention to detail from the referee. All such cases of a space missing have been found and corrected. Listed are the locations where a space needed to be added between the wavelength and unit:
p1L17;p3L21;p4L13;p6:L5,L24,L29,L31,L32;p7L10;p8L7,L29,L30;p10L2,L3,L10,L11,L12

Comment #3:
*Some values and errors are reported with too many significant digits*

Response to comment #3: The authors have carefully gone through the manuscript to confirm consistency of significant digits between values and errors within each analysis. Several cases of

excessive significant figures were found and have now been altered (P10L11,P10L13,P10L15,P10L31). This comment from the referee is much appreciated.

Comment #4:
*The optical power in these multipass cells is not unknowable. It can be determined with few simple measurements. 1) Measure the transmission of the rear mirror. 2) place a calibrated optical power meter to measure the optical power leaking through the mirror. 3) account for the mirror transmission and a factor two for a similar amount of light leak through the front mirror to get the optical power in the acoustic cavity. The issue with a fundamental calibration is that the overlap integral of the laser, acoustic mode, and aerosol is not known accurately enough for calibrations. Or possibly the microphone sensitivity and/or the laser power are not known accurately enough for calibration purposes.*

Response to comment #4:
We agree that the issues were oversimplified in the AMTD manuscript.  We have adopted to reviewer's suggestion and the passage on P3L1 now reads: "For multi-pass instruments it is difficult (Fischer and Smith, 2018b) or not feasible given the instrument setup (Lack et al., 2012b) to know all of these terms accurately. This means the first principles approach of Arnott et al. (1999) is not possible for many instruments. The issue with a fundamental calibration is that the overlap integral of the laser, acoustic mode, and aerosol is not known accurately enough for calibrations. Additionally, the microphone sensitivity and laser power are not known accurately enough for calibration purposes in the design of Lack et al. (2012b)."

Comment #5:
*P3L23: 'area' should be 'are'*

Response to comment #5: corrected

Comment #6:
*P4L11: extra space before "Lack"*

Response to comment #6: corrected

Comment #7:
*P4L19: Because this acoustic cavity consists of two high coupled resonators, there are two "primary" eigenmodes. In one mode the pressures at center of both resonators are in-phase and in the other the pressures are 180 deg. out of phase. Please rewrite the sentence to clarify which mode is being used.*

Response to comment #7: Thank you for this input, this sentence has been reworded to explain that in our instrument, the antinodes are at the center of each cell and are 180 degrees out of phase.

P4L19: "The primary eigenmode of this instrument consists of one full wavelength across the two cells, such that the antinodes are at the center of each cell and 180 degrees out of phase."

Comment #8:
*P5L4: either "each cell's resonant frequency" or "each cells' resonant frequencies" would be appropriate.*

Response to comment #8: We appreciate this attention to detail. Pertaining to both this comment and comment #9, this sentence now reads:

"To obtain the maximum and consistent signal the lasers must be modulated at each cell's resonant frequency, which is dependent on temperature and pressure in the cell. Accordingly, a resonant frequency calibration is performed at regular intervals (typically at least every 5 minutes), to account for any drifts in temperature or pressure."

Comment #9:
*P5L5: please replace "whatever interval" with a more formal phrase*

Response to comment #9: Thank you, the sentence has been corrected with "a regular interval", and we clarify that we have not set a specific timing, but this frequency calibration is done at least every 5 minutes. See above for the new sentence.

Comment #10:
P5L6: *The microphone part number does not need to be repeated here.*

Response to comment #10: corrected

Comment #11:
*P5L15: track changes indicator on this line*

Response to comment #11: corrected, thank you

Comment #12:
*P6L2-6: If known please state the concentration of solution that is used. Even if it is not critical, it is a good starting point for future projects and replication.*

Response to comment #12: The starting amounts of Nigrosin/Regal Black/Aquadag were not recorded due to the variability between tests and the need to add substances or serially dilute. However, we have added an indication of the amount of substance that was initially added and the amount of water that was initially used.

P6L2: "A few crystals (solids) or a quarter spatula (slurry), of the given substance is mixed with Milli-Q water (Millipore system SimPak2) and progressively diluted (starting with a couple hundred ml of water) until the size distribution of the atomized aerosols is such that 99% of the mass is below 300 nm. More dilute solutions tend to yield aerosol with smaller sizes."

Comment #13:
*P6L21: in this sentence please replace the 'extinction' with 'extinction channel'*

Response to comment #13: corrected

P6L30: "The scattering channel for the CAPS PM$_{SSA}$ is calibrated relative to the extinction channel, because the extinction does not require calibration."

Comment #14:
*P7L25-30: Nigrosin has a complex absorption spectrum and is not appropriately modeled with an angstrom exponent model.*

Response to comment #14: The authors agree that Nigrosin does not follow the relationship between wavelength and absorption modeled by an angstrom exponent. The discussion in section 3.2 has been modified to add additional discussion of the issues with Nigrosin. Given the small difference in wavelength between 405 and 450 nm, it is still reasonable to assess the error in the AAE method of calibrating with Nigrosin. Of the three substances, we demonstrate that Nigrosin has the least reliability for calibration, partially due to this complex relationship between absorption and wavelength. Modifications to the text begin on p7,L29:

"Additionally, the Nigrosin tested here yielded a negative AAE, which is inconsistent with figure 4 of Bluvshtein et al., (2017) in the wavelength range of 400-450 nm that shows a positive AAE. Nigrosin has been shown to have an index of refraction that significantly varies across the visible wavelengths (Bluvshtein et al., 2017), and does not have a relationship between absorption and wavelength that is appropriately modeled by AAE.. However, given that the adjustment is only over a small wavelength range (11% difference in wavelength between 450 nm and 405 nm), the error introduced by adjusting absorption measurements from 450 to 405 nm with the AAE technique is assessed here."

Comment #15:
*P8L17: Use of the Allan deviation is common in the atmospheric community where it is implicitly presented as a detection limit as a function of averaging time. The Allan deviation is useful for identifying drifts, but detection limits and instrument stability are more accurately characterized using the standard deviation which can also be presented as a function of averaging time. I suggest the authors use the standard deviation instead of the Allan deviation.*

Response to comment #15: We believe it is important to keep figure 4, the Allan deviation as a function of averaging time, in this paper so as to provide comparison between different instruments and to demonstrate the stability of the instrument. However, based on the comment, we also include a table in the supplement that demonstrates the standard deviation for each cell as a function of averaging time for 1 second, 30 second, and 60 second data. Additionally, the range of standard deviations measured for the four cells is now presented in the main body of the text. Added is Table 1 in the supplement, and the following text, beginning on P8L21:

"As an alternative noise assessment to the Allan deviation, the standard deviation of 1, 30, and 60 second average data are listed for all cells in Table S1 in the supplementary material. The 1 second data standard deviation varies between channels, from 0.01 to 0.12 Mm$^{-1}$, while at 30 seconds the range is 0.004 to 0.02 Mm$^{-1}$, and 60 seconds averaging has little change from 30 seconds."

| cell | 1 second data (Mm$^{-1}$) | 30 second average (Mm$^{-1}$) | 60 second average (Mm$^{-1}$) |
|------|---------------------------|-------------------------------|-------------------------------|
| 405 dry | .0092 | .0039 | .0035 |
| 660 dry | .0687 | .0321 | .0308 |
| 405 den | .0250 | .0311 | .0225 |
| 660 den | .1160 | .0196 | .0199 |

Comment #16:

*P9L10: Please remove one of the periods.*

Response to comment #16: corrected

Comment #17:
*P9L6: This manuscript should assess the overall measurement accuracy for ambient measurements. This section concludes that the accuracy of these calibrations is roughly +/- 6%. Is this the expected overall accuracy for ambient measurements? If yes, please state that explicitly. If no, please explain why.*

Response to comment #17:  Yes, this is the expected measurement accuracy based on the calibration. However, the measurement accuracy could also be affected by baseline drifts or abnormally large noise in various situations (aircraft, mobile, etc.).  Given this, we only state the accuracy of the calibration, not the accuracy of the actual ambient measurements, which will have to be assessed for every measurement campaign.

Comment #18:
*P10L23: please replace '3' with 'three'*

Response to comment #18: corrected, thank you

Comment #19:
*P11L4: There seem to be a few problems with some of the references. I suggest the authors look over all them carefully. P11L24: Both AMT and AMTD versions of Bluvshtein are in the reference list P12L2: "K??rcher" P12L21: no journal listed P13L19&22: Initials for Lack should capitalized. P13L31: remove "(Julie)" P15L21: No Journal listed*

Response to comment #19: thank you for this attention to detail, the full reference list has been worked over and corrected, with the specific problems addressed. The author also found an additional error and made the appropriate fix.

Comment #20:
*Figure 1: Could you re-arrange this figure so the flow goes from left to right. Maybe add arrows.*

Response to comment #20: Figure 1 has now been re-arranged so that flow goes from left to right, and arrows have been added to show the direction of flow.

Comment #21:
*Figure 2: Please change the vertical axis to 'nominal scattering' to indicate it is uncalibrated.*

Response to comment #21: The authors thank the referee for this helpful suggestion, and agree that 'nominal scattering' is more correct for this figure, given that the calibration slope has to be applied for true scattering. The title on the vertical axis of figure 2 has been changed.